# An Improved GPS-Inferred Seasonal Terrestrial Water Storage Using Terrain-Corrected Vertical Crustal Displacements Constrained by GRACE

**Hok Sum Fok [1,2,*]** and **Yongxin Liu [3,4]**

1   School of Geodesy and Geomatics, Wuhan University, Wuhan 430079, China
2   Key Laboratory of Geospace Environment and Geodesy, Ministry of Education, Wuhan University, Wuhan 430079, China
3   School of Earth and Space Sciences, Peking University, Beijing 100871, China; yxliugeo@pku.edu.cn
4   Engineering Research Center of Earth Observation and Navigation (CEON), Ministry of Education of the PRC, No. 5 Yiheyuan Road, Haidian District, Beijing 100871, China
*   Correspondence: xshhuo@sgg.whu.edu.cn; Tel.: +86-027-6877-8649

**Abstract:** Based on a geophysical model for elastic loading, the application potential of Global Positioning System (GPS) vertical crustal displacements for inverting terrestrial water storage has been demonstrated using the Tikhonov regularization and the Helmert variance component estimation since 2014. However, the GPS-inferred terrestrial water storage has larger resulting amplitudes than those inferred from satellite gravimetry (i.e., Gravity Recovery and Climate Experiment (GRACE)) and those simulated from hydrological models (e.g., Global Land Data Assimilation System (GLDAS)). We speculate that the enlarged amplitudes should be partly due to irregularly distributed GPS stations and the neglect of the terrain effect. Within southwest China, covering part of southeastern Tibet as a study region, a novel GPS-inferred terrestrial water storage approach is proposed via terrain-corrected GPS and supplementary vertical crustal displacements inferred from GRACE, serving as "virtual GPS stations" for constraining the inversion. Compared to the Tikhonov regularization and Helmert variance component estimation, we employ Akaike's Bayesian Information Criterion as an inverse method to prove the effectiveness of our solution. Our results indicate that the combined application of the terrain-corrected GPS vertical crustal displacements and supplementary GRACE spatial data constraints improves the inversion accuracy of the GPS-inferred terrestrial water storage from the Helmert variance component estimation, Tikhonov regularization, and Akaike's Bayesian Information Criterion, by 55%, 33%, and 41%, respectively, when compared to that of the GLDAS-modeled terrestrial water storage. The solution inverted with Akaike's Bayesian Information Criterion exhibits more stability regardless of the constraint conditions, when compared to those of other inferred solutions. The best Akaike's Bayesian Information Criterion inverted solution agrees well with the GLDAS-modeled one, with a root-mean-square error (RMSE) of 3.75 cm, equivalent to a 15.6% relative error, when compared to 39.4% obtained in previous studies. The remaining discrepancy might be due to the difference between GPS and GRACE in sensing different surface water storage components, the remaining effect of the water storage changes in rivers and reservoirs, and the internal error in the geophysical model for elastic loading.

**Keywords:** terrestrial water storage inversion; GPS; GRACE; GLDAS; Akaike's Bayesian Information Criterion; virtual GPS stations

## 1. Introduction

Terrestrial water storage (TWS), being one of the indispensable hydrological components, consists of surface water storage (e.g., reservoir and lake), groundwater, soil moisture, snow, and ice [1]. Its spatial redistributions cause hydrological loading that deforms the solid Earth, in which vertical crustal displacements (VCD) are of major interest in geophysics and geodesy [2]. Due to the spatially sparse and point-wise *in situ* measurements, along with the labor-intensive acquisition process [3], the regional and global TWS variations are less known when compared to other water balance components, such as precipitation and discharge.

Recent advances in space geodetic sensors, such as Gravity Recovery and Climate Experiment (GRACE) satellite gravimetry and global positioning system (GPS), make global and evenly distributed TWS data available at basin-wide, regional, and global scales (e.g., [4,5]). Given the first discovery of seasonal signals in GPS VCD [6] and the preliminary TWS (expressed in terms of the equivalent water height (EWH)) inferred by GRACE since 2000 (e.g., [4,7]), GPS (e.g., [8,9]) or GPS in combination with GRACE (e.g., [2,10–12]) has been widely employed to monitor the seasonal surface deformation, including hydrological extremes [13]. By converting the monthly GRACE-inferred TWS into VCD (hereinafter denoted as GRACE VCD), the GRACE VCD is quantitatively comparable to that of GPS [11].

More recently, a geophysical observation model was set up via the well-known Green's function [14] for converting the seasonal GPS VCD into TWS (hereinafter denoted as GPS-inferred TWS) with the Laplacian smoothness constraint [5]. Using the GPS Plate Boundary Observatory (PBO) network in California, the resulting GPS-inferred TWS was qualitatively comparable to the GRACE-inferred TWS and hydrological models, such as the Global Land Data Assimilation System (GLDAS) [15,16]. Jin and Zhang (2016) [1] applied the same method to drought monitoring in the entire southwestern region of the United States. In addition, a Helmert variance component estimation (HVCE) re-weighting technique was applied for the inversion, along with a critical quantitative assessment [17].

In the abovementioned study, the GPS-inferred TWS yields larger amplitudes than those of the GRACE-inferred and GLDAS-modeled TWS. We speculate that this is probably attributable to the sparsely uneven distribution of the GPS stations that provide less spatial data constraints in remote areas, leading to poor results when compared to the GRACE-inferred and GLDAS-modeled TWS. In addition, the well-known geophysical observation model, relating the contribution of loading masses to VCD, assumes an elastic and ideal rigid plane surface. However, most regions with a substantial seasonal water storage are located in the upstream area of the river basin, where a sloped surface and an irregular terrain are presented. These regions contradict the plane surface assumption. These two situations necessitate further spatial data constraints and terrain-corrected VCD in the conversion process.

The conversion process from the seasonal GPS VCD to the GPS TWS is fundamentally an ill-posed inverse problem, because the number of station observations is far fewer than the number of gridded parameters to be determined. This conversion process requires regularization (e.g., [18]), for which a regularization parameter (equivalent to a weighting ratio) that balances the contribution between the observations and constraints is sought. Various methods have been proposed to determine the regularization parameter, including HVCE (e.g., [19–21]), the generalized cross-validation (GCV) (e.g., [22–24]), Akaike's Bayesian Information criterion (ABIC) [25], MMSE criterion [26], and the L-curve method [27,28]. Although L-curve [15] and HVCE [16] have been demonstrated in previous studies, these methods are subjective in terms of model selection and computationally expensive for correlated data. In addition, HVCE is sensitive to an initial value that might cause its divergence.

ABIC, the hybrid of the Akaike's information criterion (AIC) and the Bayesian information criterion (BIC), is a widely used objective method based on the maximization of entropy. It aims to minimize the prediction error while preserving the consistency of probability in the model selection. Recent advances in geophysical data inversion that determines the regularization parameter tend to be in favor of the ABIC method. It has been applied in the finite fault slip distribution inversion [29],

in the waveform record inversion of rupture processes with both spatial and temporal smoothness constraints (e.g., [30]), and in the joint inversion of rupture processes using geodetic observations and seismological waveforms (e.g., [31]). Its generalized form for different kinds of datasets has also been formulated [32]. Therefore, it is worthwhile investigating its application in the conversion of seasonal GPS VCD into GPS-inferred TWS, as the conversion basically belongs to a kind of geophysical data inversion.

Using 34 continuous GPS stations from the Crustal Movement Observation Network of China (CMONOC) in southwest China, this study aims to demonstrate a substantial inversion improvement in estimating the seasonal TWS from GPS VCD via ABIC, with the considerations of spatial data constraints and terrain-corrected VCD. Supplementary GRACE VCD are employed as "virtual GPS stations" for a priori information within the study region, and GPS VCD are corrected for the terrain effect during the inversion process. The proposed methodology is then internally compared to that of the Tikhonov regularization (TR) and HVCE methods, while it is externally compared to the GLDAS-modeled TWS.

## 2. Study Region and Data Description

### 2.1. Southwest China

Southwest China (Figure 1), covering the entire Yunnan and Guizhou provinces and part of the southeastern Tibetan Plateau, is situated at the upstream area of the Lancang River (also called Mekong River outside China), Yangtze River, and Pearl River. This region is climate-driven and affected by the Indian monsoon [33]. Its altitude ranges from 1500 to 4000 m. A bare karst geology is located at the eastern part of the study region, with low-permeability rocks surrounded by high-permeability ones. This causes water to infiltrate quickly into the ground, which results in a surface water deficit [31]. In addition, half a billion people live in the mid- and downstream areas of the above river basins, which are vulnerable to hydrological extremes occurring in the upstream area [34–36]. The hydrological extremes in the upstream area threaten the agriculture, living environments, and the economy in the mid- and the downstream areas whether within China or her transboundary river basins. This indicates that the large-scale monitoring of TWS is essential for sustainable agricultural practices, water supply, and for the notification of extreme events within the region. The geographic environment and its apparent seasonal characteristics (as manifested from the apparent seasonal signals in GPS VCD in Figure 8 of [37]) in this region lay out a good environmental setting for the study.

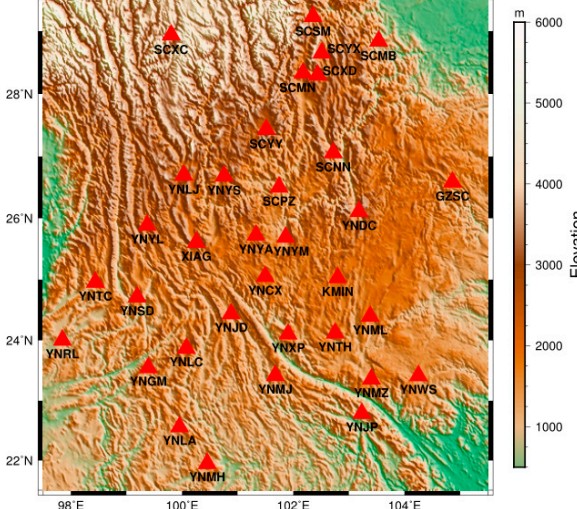

**Figure 1.** Topographic variations of southwest China with continuous global positioning system (GPS) station locations (in red).

### 2.2. GPS VCD Data

The frequent seismic activities and other geological hazards necessitate the installation of 34 continuous GPS observation stations in southwest China, affiliated with CMONOC of China (Figure 1) [38]. The GPS VCD data could be accessed via http://www.cgps.ac.cn/. Most continuous GPS stations in the study region have been initiated since 2010, except the KMIN and XIAG stations, which have been operated since 1999. The variable trends and gross errors in VCD were presented for the first two years of observations. Therefore, the time series of all GPS stations spanning between January 2012 and October 2014 were utilized in this study.

### 2.3. GRACE and GLDAS TWS Data

GRACE is a satellite mission project of NASA and the German Aerospace Center (DLR); it measures time-variable Earth's gravity changes that can be converted into TWS variations on a global scale [4]. The degree-90 GRACE Level-2 Release 05 (RL05) GSM monthly gravity data product, released by JPL, contains spherical harmonic coefficients (i.e., Stokes coefficients) that represents the mass variations [39,40]. These data are freely available at https://grace.jpl.nasa.gov/. This product allows us to calculate the time series of TWS (in terms of EWH in cm at a regular grid) using Equation (14) in [7], divided by the water density. A Gaussian filter at a radius of 300 km [41] and DDK decorrelation filter [42] are applied to reduce the spatially correlated error of the TWS data at higher degrees, with a final gridded TWS at a $1° \times 1°$ grid. Subsequently, GRACE TWS are converted into GRACE VCD by the forward modeling formulation in [11], in which the load Love numbers are adopted from [43]. This is followed by introducing GRACE VCD at specified locations as "virtual GPS stations", serving as spatial data constraints for the inversion process. Note that no ocean tidal loading correction is applied due to the study region being located far away from the coast.

GLDAS is a system that assimilates both the satellite and ground-based observational data products to generate optimal hydrological data [16]; it has four land surface models: the Community Land Model (CLM), Variable Infiltration Capacity (VIC), Mosaic, and the national centers for environmental prediction/Oregon State University/Air Force/Hydrologic research lab (NOAH). The NOAH model is adopted in this study, for which the modeled data are available at https://mirador.gsfc.nasa.gov/. By the summation of the canopy water, soil water, and snow data layers, the GLDAS-modeled TWS is calculated. The GLDAS-modeled TWS data were served as external data for comparison with the GPS-inferred TWS resulting from the proposed methodology and the existing ones. The GRACE and GLDAS data time spans were identical to that of GPS.

## 3. Data Processing of GPS Time Series

### 3.1. GPS Data Preprocessing

The GAMIT version 10.4 software [44] was employed for preprocessing the CMONOC GPS observation data to generate a daily solution of station coordinates and their uncertainties. By processing 34 CMONOC GPS stations together with 24 International GNSS Service (IGS) [45] stations surrounding China, the network solution for the coordinates of the 34 CMONOC GPS station time series was aligned to the International Terrestrial Reference Frame 2008 (ITRF2008) [46]. The uncertainties of the 34 CMONOC GPS station coordinates were initially constrained to 100 m by using approximate coordinates from a point-position solution, whereas those of the 24 IGS station coordinates were constrained to 5 cm. The Earth Orientation Parameters (EOP) were also constrained to a priori values listed in the International Earth Rotation Service (IERS) Bulletin B.

During the preprocessing, corrections were applied in GAMIT to remove systematic effects. The following correction procedures were applied: (i) to fix the precise orbits by using the IGS final ephemeris products; (ii) to correct for the ionospheric delay by up to three order terms [47] in GAMIT; (iii) to correct for the tropospheric delay by using the Vienna mapping function 1 (VMF1) [48] with a cut-off value of $10°$ and a priori hydrostatic delays provided by the global pressure and temperature

(GPT) model [49]; (iv) to correct for the receiver antenna offsets using the IGS antenna correction files; (v) to correct for the non-tidal atmospheric loading by the MIT atmdisp_cm.year files; (vi) to correct for the ocean tidal loading by choosing the FES2004 model [50] option in GAMIT [44], even though the study region is not close to the coast; and (vii) to correct for the solid Earth tides and pole tide according to the IERS standard [51].

Environmental loading includes atmospheric, hydrological, and non-tidal ocean loading (NTOL) [52]. Since the hydrological loading signal is our main concern, other environmental loading signals in the GPS VCD time series should be corrected. Note that the atmospheric loading has been corrected in the aforementioned preprocessing. Therefore, GPS VCD induced by NTOL should be corrected to obtain a purely hydrological loading signal due to TWS. This can be corrected by using the half-daily modeled NTOL displacement data with a 2.5° × 2.5° spatial resolution obtained from Global Geophysical Fluid Center (GGFC); these data are available at http://geophy.uni.lu/.

### 3.2. GPS Data Post-Processing

After the pre-processing, outliers larger than twice the standard deviation were identified and removed to enhance the data quality. However, before estimating the seasonal signals from the GPS, aliasing and draconitic errors (i.e., ~351 days [53,54]), which lead to a biased estimation of amplitudes at different frequencies, have to be mitigated [55,56]. Therefore, the GPS station time series were transformed into power spectra via a Fast Fourier transformation (FFT) for further investigation (Figure 2). The 1st peak dominant period was observed at ~318.3 days, departed from the annual signal (i.e., 1 cycle per year (cpy)), while the 2nd peak varied irregularly at different frequencies that deviated significantly from the true semi-annual period (i.e., 2 cpy). Since the annual variation was the desired seasonal signal, filtering was applied in the frequency domain to reduce the influence of the aliasing and draconitic errors on the annual signal. After filtering in the frequency domain, the 1st peak is apparently closer to 1 cpy. Each filtered GPS spectrum was then transformed back into its corresponding time series via the inverse FFT. To demonstrate the differences, the unfiltered and filtered time series are displayed in Figure 3. In general, the peaks and troughs of the GPS time series are reduced after filtering in the frequency domain.

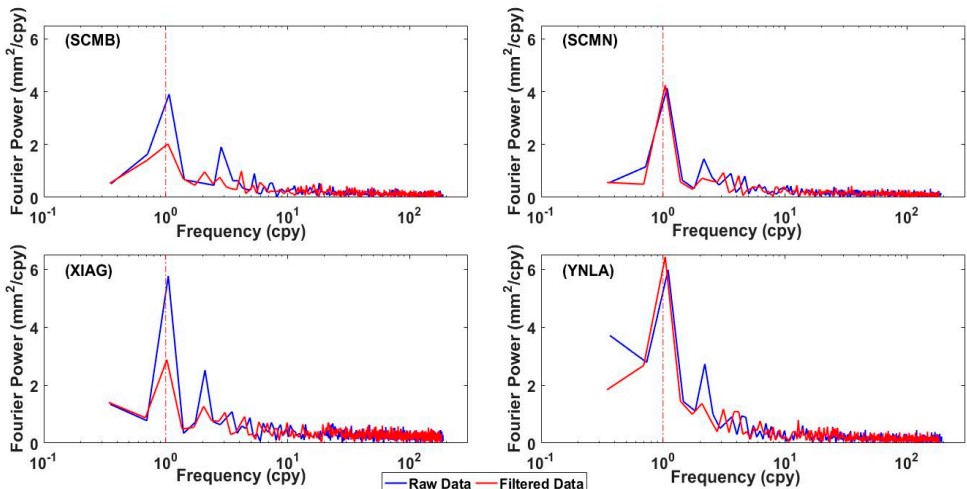

**Figure 2.** The GPS Fourier power spectra of the unfiltered (blue) and filtered (red) time series for four selected GPS stations (i.e., SCMB, SCMN, XIAG, and YNLA).

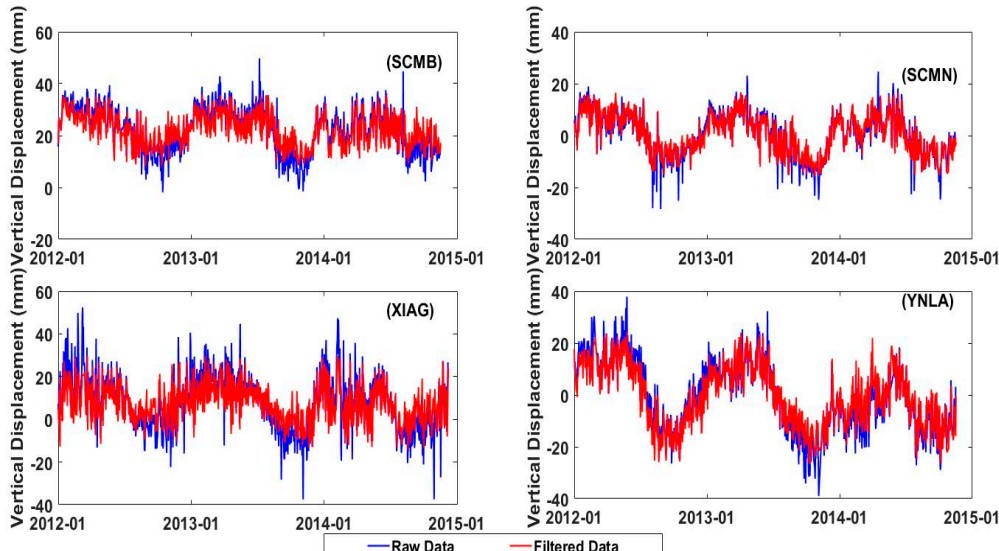

**Figure 3.** The comparison of the GPS height time series before (green) and after (red) filtering in the spectrum domain for four selected GPS stations (i.e., SCMB, SCMN, XIAG, and YNLA).

To decorrelate the time series of different GPS stations [57] and to enhance the signal-to-noise ratio [58], the spatially-correlated common mode error (CME), first discovered in [59], has to be eliminated via the regional stacking filtering technique. This is achieved by calculating the regional offset from averaging the residual time series of all GPS stations within the region, followed by removing it from each GPS time series [60]. This procedure can further improve the signal-to-noise ratio that sharpens the main signal in the GPS time series (Figure 4).

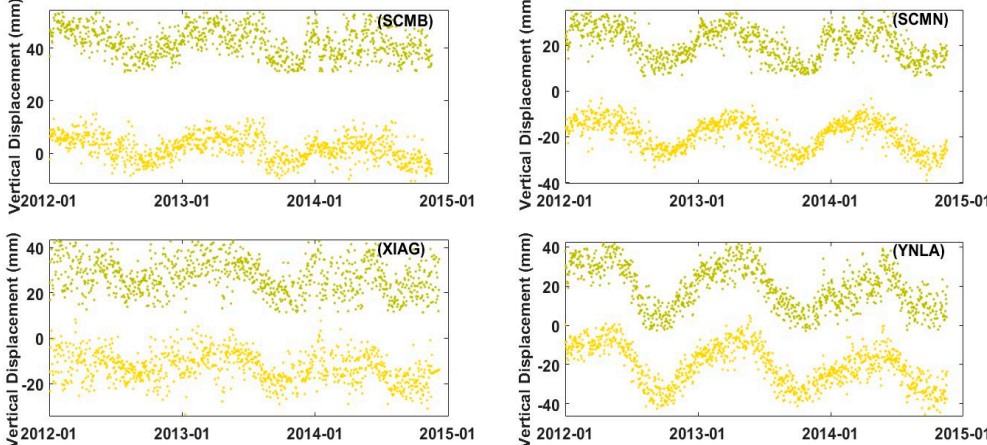

**Figure 4.** GPS time series before (green) and after (yellow) regional stacking filtering at selected stations (with station name in the top-right corner of each time series).

To estimate the seasonal signal, a linear model, including the offset, trend, and periodic terms, was adopted as:

$$y = a + b(t - t_0) + \sum_{i=1}^{2} A_i \cos\left(\frac{2\pi}{T_i}(t - t_0) + \varphi_i\right) + e \qquad (1)$$

where $a$ and $b$ denote the offset and the linear trend, respectively; $A_i$ and $\varphi_i$ denote the amplitude and phase of a seasonal signal $i$ with a period $T_i$ to be estimated, respectively; and $e$ denotes the error term that contains colored noise (e.g., random walk and flicker noise) in the GPS data time series [60]. The initial observation time epoch, $t_0$, is fixed to January 1, 2012. Both $T_1 = 1$ and $T_2 = 0.5$ are the annual

and semiannual periods, respectively. Equation (1) is also applied to the GRACE and GLDAS TWS data time series.

Due to the presence of the colored noise, the nearly 3-year data span is too short for a proper recovery of the annual signal. The traditional least-squares method will also lead to a biased estimation [61]. Consequently, a maximum likelihood estimation was employed to estimate all of the unknown parameters in Equation (1), as the colored noise models can be assumed and estimated in order to better recover the annual signal. The Hector software package [62] employing a maximum likelihood estimation was used to determine all parameters. The power-law noise and white noise combination was chosen as the noise model algorithm within the Hector software package option. Therefore, the annual amplitude, representing the dominant seasonal signal, was estimated for each GPS site for this study. Furthermore, the annual amplitude becomes larger when the annual variation of the land surface temperature increases [63]. In our study region, this annual variation contributes to an upward bedrock deformation of about 0.5 mm, equivalent to a ~1 cm amplification of TWS, as tested. Therefore, the estimated annual amplitudes of the GPS were corrected for a bedrock thermal expansion, according to [64].

Although the annual amplitude of all the GPS stations manifested a general decrease from the southwest to northeast of the study region (Figure 5a), a site-dependent distribution was observed. This can be attributed to the GPS being sensitive to near-field or local variations of hydrological loading. A different elastic response resulting from the local crustal structure [65] or other unmodeled geophysical processes [66] can be a reason. Due to an insufficiency of GPS stations, stations with a relatively high standard deviation have not been eliminated (Figure 5b).

Instead of a planar surface, the actual hydrological loading exerts its force on the slope surface, which contradicts the rigid plane assumption in the loading theory [14]. While it can be negligible for plain and interior plateaus with a small terrain variation, our study region is a mountainous area with various slopes. To account for the slope surface, a terrain correction should be applied, by multiplying each estimated GPS annual amplitude, $A_1$, with a cosine of an inclination angle, $\beta$, to yield the terrain-corrected GPS annual amplitude, $u$, as:

$$u = A_1 \cos(\beta) \tag{2}$$

According to the characteristics of the topography variation in our study region, we equally divide the study region into four sub-regions, where the inclination angles of 31.1°, 15.5°, 18.8°, and 19.5° are applied in the northwest, northeast, southwest, and southeast sub-regions, respectively.

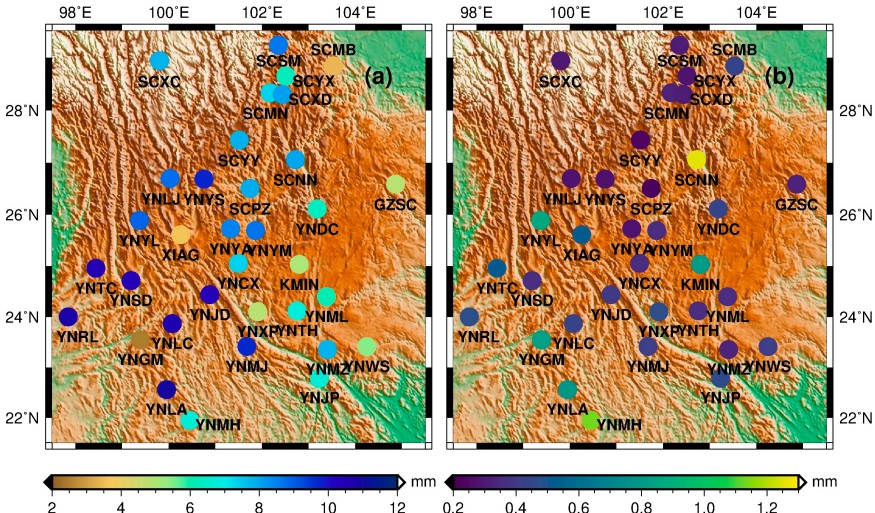

**Figure 5.** Distribution of (**a**) the annual amplitude and (**b**) its standard deviation (in mm) of the 34 GPS stations on a topographic base map.

## 4. TWS Inversion Model and Methodology from an Annual Variation of GPS VCD

The field displacement at a point, as observed from a GPS station, represents the response of the all loading masses contribution on the rigid plane surface with an elasticity assumption [14]. The displacement, *du*, due to the contribution of a point mass at an arbitrary point, is formulated as the product of the Green function and a point mass, *dm*, given as:

$$du = G(\theta)dm,$$ (3)

where $G(\theta)$ is the Green's function with the angular distance $\theta$ between the point mass, *dm*, and the field displacement point (i.e., the GPS station location). The field displacement at a point resulting from all of the loading masses, *u*, requires the summation of all the loading masses on the surface. Although all of the loading masses on the Earth's surface should be accounted for in theory, the surface integral region is limited to an extra 5° in each direction outside the study region in practice due to the negligible contribution of the distant mass load and a reduction in the computation workload.

Since there is a limited number of GPS stations in our study region, the observed *u* values are far fewer than the parameters $\xi$ (i.e., the TWS of each $1° \times 1°$ grid in the study region with the extra 5° in each direction that is to be estimated). Consequently, this is an underdetermined problem which requires a regularization technique for the inversion. Tikhonov regularization (TR) is the technique employed for this purpose [18]. Thus, the linear model with constraints is given by [15]:

$$\begin{cases} A\xi = \mathbf{u} \\ L\xi = 0 \end{cases},$$ (4)

where $\xi$ and $\mathbf{u}$ denote the column vector of the TWS of each grid to be estimated and the terrain-corrected vertical annual amplitude of each GPS site, respectively. The Tikhonov matrix, *L*, is expressed in terms of the Laplacian operator ($\nabla^2$) for smoothing the solutions in this study. The design matrix, *A*, is expressed as:

$$A = \begin{pmatrix} \iint_{\Omega_1} \rho_w G(\theta_1)dS & \iint_{\Omega_2} \rho_w G(\theta_1)dS & \cdots & \iint_{\Omega_i} \rho_w G(\theta_1)dS \\ \iint_{\Omega_1} \rho_w G(\theta_2)dS & \iint_{\Omega_2} \rho_w G(\theta_2)dS & \cdots & \iint_{\Omega_i} \rho_w G(\theta_2)dS \\ \vdots & \vdots & \ddots & \vdots \\ \iint_{\Omega_1} \rho_w G(\theta_j)dS & \iint_{\Omega_2} \rho_w G(\theta_j)dS & \cdots & \iint_{\Omega_i} \rho_w G(\theta_j)dS \end{pmatrix},$$ (5)

where $\rho_w$ is the water density; $\Omega_i$ is the integral surface patch $i$, $\theta_j$ is the angular distance between a point mass within $\Omega_i$ and the field displacement point $j$ (i.e., the location of the GPS site $j$); and *dS* is the surface area of each mass grid. Thus, the regularized solution of the estimated GPS-inferred TWS with equal weight from Equation (4) is given by:

$$\widehat{\xi} = (A^T A + kL^T L)^{-1} A^T \mathbf{u},$$ (6)

where *k* is the regularization parameter. While TR determines the value of *k* by obtaining the minimum of the trade-off curve between the model fitness and *k*, HVCE optimizes among the observations and constraints via their respective variance components iteratively. In this study, the equations for HVCE to determine the respective variance components can be written as [21]:

$$H\hat{\gamma} = q$$ (7)

where $H = \begin{bmatrix} n_1 - 2tr(N^{-1}N_1) + tr(N^{-1}N_1N^{-1}N_1) & tr(N^{-1}N_1N^{-1}N_2) \\ tr(N^{-1}N_1N^{-1}N_2) & n_2 - 2tr(N^{-1}N_2) + tr(N^{-1}N_2N^{-1}N_2) \end{bmatrix}$, $\hat{\gamma} = \begin{bmatrix} \hat{\sigma}_{01}^2 & \hat{\sigma}_{02}^2 \end{bmatrix}^T$, $q = \begin{bmatrix} \mathbf{e}_1^\mathbf{T}\mathbf{e}_1 \\ k\mathbf{e}_2^\mathbf{T}\mathbf{e}_2 \end{bmatrix}$, $N = N_1 + N_2$, $N_1 = A^T A$, $N_2 = kL^T L$, $\mathbf{e}_1 = \mathbf{u} - A\xi$, $\mathbf{e}_2 = L\xi$, where $n_1$ and $n_2$ are the numbers of the observation and constraint equations, respectively; $\hat{\sigma}_{01}^2$ and $\hat{\sigma}_{02}^2$ are the

estimated variance components for the observation and constraint equations to be solved iteratively, respectively; and $\mathbf{e}_1$ and $\mathbf{e}_2$ are the column vectors of the observation and constraint residuals, respectively. By initializing a small value of $k$ (i.e., $1 \times 10^{-4}$), the value of $k$ is iteratively determined via:

$$k_l = \frac{\hat{\sigma}^2_{01,l-1}}{\hat{\sigma}^2_{02,l-1}} k_{l-1}, \tag{8}$$

until $\hat{\sigma}^2_{01}$ and $\hat{\sigma}^2_{02}$ are unchanged in the in the $l$-th iteration.

Akaike's Bayesian information criterion (ABIC) is another technique for determining the regularization parameter based on the maximization of entropy. The Bayesian theorem can describe the TWS probability distribution in our study region for given observations (i.e., GPS) and prior constraints. Instead of choosing the regularization parameter $k$ by the trade-off curve or variance components iteratively, ABIC can search for the parameter $\alpha^2$ that minimizes the ABIC value as [31,32]:

$$ABIC(\alpha^2) = n \log(s(\boldsymbol{\xi}^*)) - log|\alpha^2 L^T L| + log|A^T A + \alpha^2 L^T L| + C' \tag{9}$$

where $n$ is the number of observations; $\alpha^2$ is the smoothness parameter, equivalent to the above parameter $k$; and $C'$ is a constant term. Note that an equal weight of each observation is applied, and thus no covariance matrix of the observation vector is presented in the regularized solution formulation. $F(\boldsymbol{\xi})$ is the Lagrangian target function, equivalent to Equation (4), shown as follows:

$$F(\boldsymbol{\xi}) = (\mathbf{u} - A\boldsymbol{\xi})^T (\mathbf{u} - A\boldsymbol{\xi}) + \alpha^2 \boldsymbol{\xi}^T L^T L \boldsymbol{\xi}, \tag{10}$$

with the regularized solution of the estimated GPS-inferred TWS with equal weights solved by:

$$\boldsymbol{\xi}^* = (A^T A + \alpha^2 L^T L)^{-1} A^T \mathbf{u}, \tag{11}$$

which is, indeed, identical to Equation (6), except that $\alpha^2$ is determined by the minimum ABIC value.

Due to the uneven distribution of the GPS stations with fewer stations near the boundary of the study region, a poor GPS-inferred TWS is expected. To remedy the situation, GRACE VCD, serving as "virtual GPS stations", are introduced at four locations close to the boundary of the study region that follows the southeast descending trend of the terrain with poor coverage of the GPS stations. Three more virtual GPS stations are chosen to compensate for the sparse GPS situated on the steep slope (i.e., inclination angles of 31.1°) in the northwest of the study region (Figure 6). Note that the GRACE VCD can be converted by the forward modeling formulation in [11] using GRACE-inferred TWS, in which the load Love numbers are adopted from [43]. The annual amplitudes of the GRACE VCD are subsequently synthesized using Equation (1) for the pre-defined virtual GPS stations based on the above criteria. The chosen virtual GPS stations should be as few as possible, so that the GPS can dominate the inversion process. This strategy is equivalent to the introduction of stochastic constraints into the linear observation model [67].

Thus, Equation (4) can be extended to:

$$\begin{cases} A\boldsymbol{\xi} = \mathbf{u} \\ L\boldsymbol{\xi} = 0 \\ A_G\boldsymbol{\xi} = \mathbf{u}_G \end{cases}, \tag{12}$$

where $\mathbf{u}_G$ denotes the GRACE VCD of the specified site locations, and $A_G$ is the corresponding observation matrix with the size depending on the number of synthetic virtual GPS stations from the GRACE VCD. The regularized solution of Equation (12) can be expressed as:

$$\widehat{\boldsymbol{\xi}} = (A^T A + k L^T L + A_G{}^T A_G)^{-1} (A^T \mathbf{u} + A_G{}^T \mathbf{u}_G). \tag{13}$$

By ignoring the potential covariances existing between GPS VCD and additional constraints, the ABIC regularized solution can be simplified by replacing $\mathbf{u}_1 = \begin{pmatrix} \mathbf{u} & \mathbf{u}_G \end{pmatrix}^T$ and $A_1 = \begin{pmatrix} A & A_G \end{pmatrix}^T$. Thus, the determination of the smoothness factor $k$ is then the trade-off among the model residuals, smoothness, and residuals of the additional observations.

## 5. Results and Discussion

To demonstrate the effectiveness of the constraint strategy and terrain correction, the TR, HVCE, and ABIC methods were employed. Subsequently, they were compared with those results without constraint and terrain correction. Regardless of the regularization methods that were employed, the GPS-inferred TWS without constraint exhibits the north-south stripping pattern with decreasing TWS amplitudes from the west to the east (Figure 6a,d,g). The TWS inverted with HVCE is generally larger than that inverted with TR or ABIC, particularly in the west of our study region.

After introducing spatial GRACE data constraint points for the joint inversion according to the above criteria, the spatial pattern of all GPS-inferred TWS exhibits similarities with the GRACE-inferred and GLDAS-modeled TWS, showing a pattern of increasing TWS amplitudes from the northwest to the southeast, which is attributed to the regional water storage characteristics in the study region (Figure 6b,e,h). The GPS-inferred TWS with spatial data constraints, as well as the terrain correction, further strengthens the consistency of the TWS spatial pattern (Figure 6c,f,i) when compared to the cases with the GRACE-inferred and GLDAS-modeled TWS (Figure 7), indicating that the introduction of virtual GPS stations along with the terrain consideration is critical for the inversion process.

To evaluate the optimal solution among the inversion approaches, the GPS-inferred TWS solutions are evaluated internally based on the model fitness and sensitivity, and are externally based on a comparison with the external observations. The root-mean-squares residuals (RMSR) of the observation equations reflect the model fitness. The RMSR closest to 1 indicate the most probable gridded TWS estimate in terms of the model fitness. While the terrain correction yields a negligible effect on the RMSR, the additional spatial GRACE data constraint points exert a relatively significant effect on the RMSR when compared to that without constraint, thereby decreasing the model fitness (Table 1). A comparison among the inversion methods reveals that the solution inverted with HVCE yields the best model fitness. However, this does not imply that the solution inverted with HVCE is superior to those of ABIC and TR, since different Lagrangian target functions have been employed, resulting in different model fitness.

While HVCE adjusts the balance among the observations and constraints via an iterative estimation of the variance components [21], ABIC optimizes the balance among them according to the information entropy [30]. TR yields the tradeoff with an equal distance between the model misfit and the roughness in the error space. Compared with HVCE and TR, ABIC displays the least changes in RMSR (Table 1), because the determined smoothness factors are close to each other in every constraint condition (Figure 6 and Table 2). This is attributable to the determination of parameter $\alpha$ based on Equation (9), which takes less account of the roughness while placing more weight on the observation model and the constraints.

**Table 1.** Statistics of the GPS-inferred TWS from various constraints (i.e., the spatial GRACE data constraints (SGDC) and terrain-corrected GPS (TC)) and inversion approaches. RMSR[1] is the variance component of the observations.

|  | RMS Error against GRACE (cm) | RMS Error against GLDAS (cm) | Mean Uncertainty (cm) | RMSR[1] |
|---|---|---|---|---|
| HVCE | 7.76 | 8.92 | 0.46 | 1.08 |
| HVCE with TC | 7.13 | 8.27 | 0.42 | 1.04 |
| HVCE with SGDC | 3.50 | 4.89 | 0.13 | 1.45 |
| HVCE with SGDC and TC | 2.60 | 4.02 | 0.12 | 1.36 |
| TR | 4.63 | 6.11 | 0.42 | 1.44 |
| TR with TC | 3.53 | 5.05 | 0.29 | 1.41 |
| TR with SGDC | 3.17 | 4.73 | 0.14 | 2.00 |
| TR with SGDC and TC | 2.45 | 4.09 | 0.14 | 1.92 |
| ABIC | 5.14 | 6.35 | 0.43 | 1.90 |
| ABIC with TC | 3.76 | 5.01 | 0.41 | 1.83 |
| ABIC SGDC | 3.51 | 4.82 | 0.34 | 1.94 |
| ABIC SGDC and TC | 2.48 | 3.75 | 0.34 | 1.84 |

To evaluate the sensitivity to changes in the unknown gridded estimates, a bootstrapping method is performed 34 times with one GPS site being eliminated each time to generate 34 solutions with each approach. Subsequently, the standard deviation of the gridded estimates is computed to determine the uncertainty (Table 1). This demonstrates that the additional spatial constraint points can reduce the overall uncertainties of the gridded estimates.

To evaluate the GPS-inferred TWS solutions externally, the GLDAS-modeled TWS are served as the external data for the accuracy assessment. Regardless of the different inversion methods, the application of the spatial GRACE data constraints reduces the root-mean-square error (RMSE) when compared to that of no constraints (Table 1), as does the terrain correction. The combined application of the spatial GRACE data constraints and the terrain correction can further reduce the RMSE against the GLDAS-modeled TWS by 55%, 33%, and 41% for HVCE, TR, and ABIC, respectively, as is also revealed and visualized in Figure 8. The results are also shown to be better than those of [17], although shorter data time spans were used in this study.

**Table 2.** Value of the regularization parameter *k* for different inversion methods with various constraints (i.e., the spatial GRACE data constraints (SGDC) and terrain-corrected GPS (TC)).

|  | HVCE | TR | ABIC |
|---|---|---|---|
| No constraint | $8.42 \times 10^{-4}$ | $1.16 \times 10^{-4}$ | $1.53 \times 10^{-4}$ |
| TC | $8.42 \times 10^{-4}$ | $1.16 \times 10^{-4}$ | $1.77 \times 10^{-4}$ |
| SGDC | $3.39 \times 10^{-4}$ | $1.80 \times 10^{-4}$ | $1.65 \times 10^{-4}$ |
| SGDC and TC | $3.47 \times 10^{-4}$ | $4.20 \times 10^{-4}$ | $1.48 \times 10^{-4}$ |

Overall, among the different GPS-inferred TWS solutions, the solution inverted with ABIC yields a consistent and stable performance, when evaluating the model fitness and sensitivity internally, and when comparing it with the GLDAS-inferred TWS externally, regardless of the presence or absence of constraints.

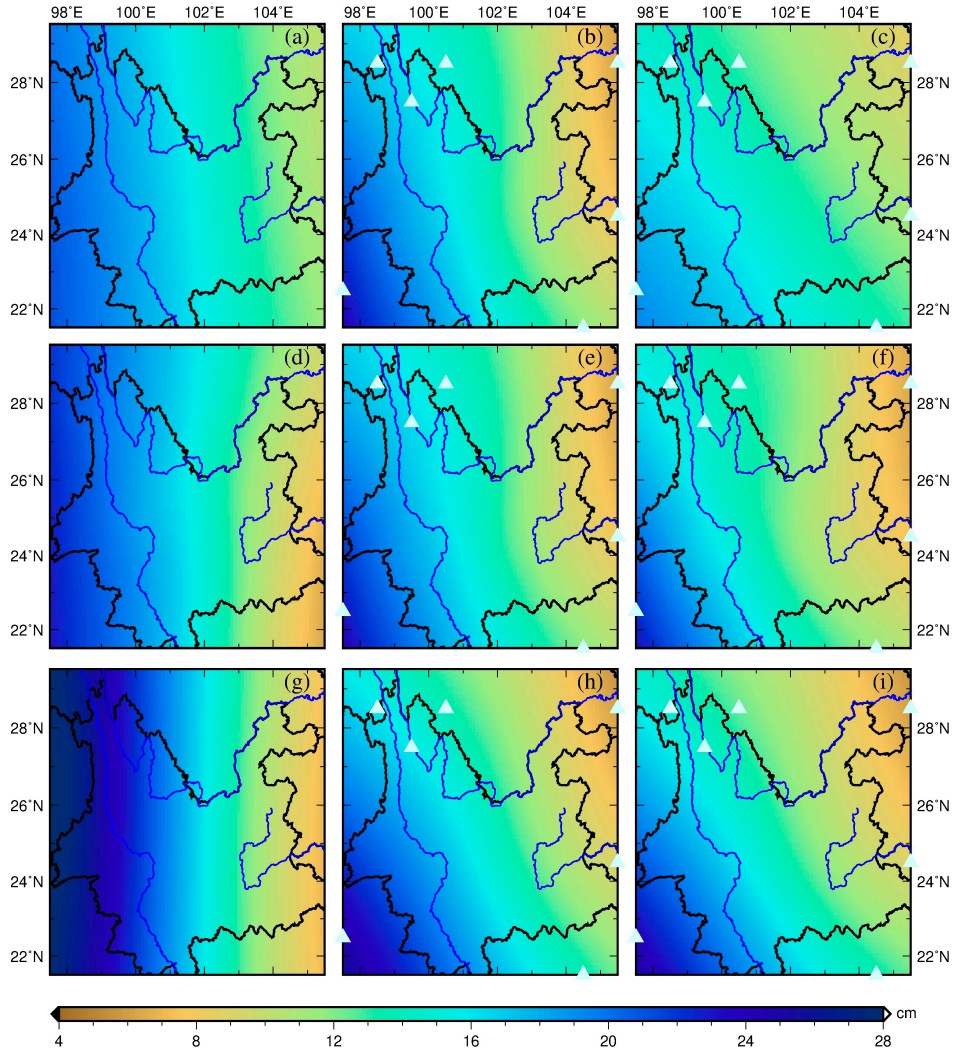

**Figure 6.** GPS-inferred TWS inverted by TR, ABIC, and HVCE, respectively, (**a**,**d**,**g**) without constraints, (**b**,**e**,**h**) with spatial GRACE data constraints and (**c**,**f**,**i**) with spatial GRACE data constraints and terrain-corrected GPS. Virtual GPS station locations (with white triangles).

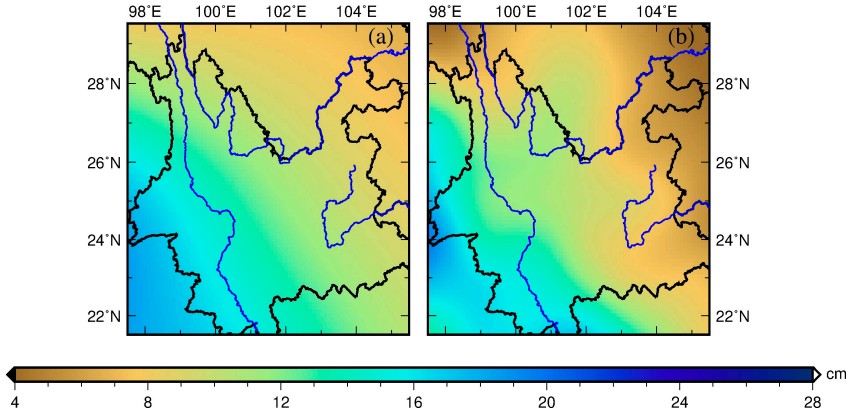

**Figure 7.** TWS (**a**) inferred from GRACE and (**b**) modeled from GLDAS.

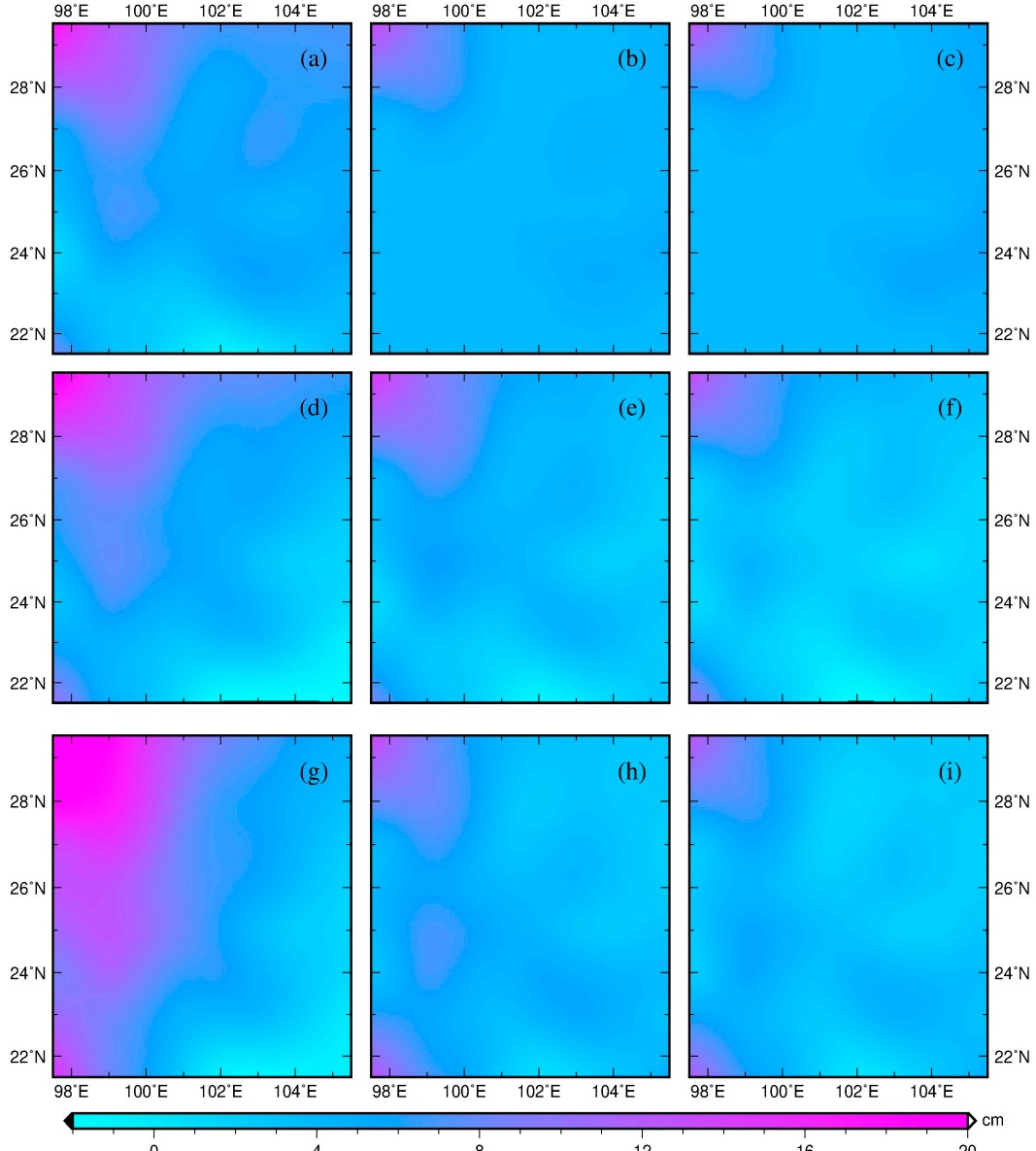

**Figure 8.** The bias of GPS-inferred TWS inverted by TR, ABIC, and HVCE, respectively, (**a,d,g**) without constraints, (**b,e,h**) with spatial GRACE data constraints and (**c,f,i**) with spatial GRACE data constraints and terrain-corrected GPS, respectively, against GLDAS-modeled TWS.

## 6. Conclusions

Contrary to the published results without spatial data constraints and terrain considerations [5,17], GPS-inferred seasonal TWS, which are improved by introducing the spatial GRACE data constraints as virtual GPS stations and by correcting the GPS VCD for the terrain effect, have been illustrated via the Helmert variance component estimation (HVCE), Tikhonov Regularization (TR), and Akaike's Bayesian Information Criterion (ABIC) inversion methods using continuous CMONOC GPS stations in southwest China. The GLDAS-modeled TWS were employed as external data for a comparison with the GPS-inferred TWS resulting from the proposed approaches and the existing ones.

It was found that the application of either the spatial GRACE data constraints or terrain correction could substantially improve the inversion quality of the GPS-Inferred TWS, despite the reduction in the model fitness quality. The combined application of the spatial GRACE data constraints and the terrain correction could further increase the accuracy of the GPS-Inferred TWS by 55%, 33%, and 41%

for HVCE, TR, and ABIC, respectively, when externally compared against the GLDAS-modeled TWS. Among the different GPS-inferred TWS solutions, the solution inverted with ABIC yielded a consistent and stable performance, regardless of the constraint conditions. The best ABIC inverted solution was achieved with a RMSE of 3.75 cm against the GLDAS-modeled TWS, equivalent to a 15.6% relative error normalized by the amplitude range.

The remaining discrepancy can be potentially attributed to the missing account of the water storage change in rivers and reservoirs, the difference between GPS and GRACE in terms of the sensitivity to surface water storage, and the internal error in the employed geophysical model. Since the inversion is fundamentally an optimization process, different weights can be assigned to the observations, the Laplacian smoothness constraint, and the spatial GRACE data constraints, for the potential improvement of the GPS-inferred seasonal terrestrial water storage.

Since different regions around the globe exhibit different hydro-climatic and topographic conditions, further experiments in regions with relatively small seasonal TWS variations and various terrain types should be conducted to confirm the feasibility of the presented methodology. Furthermore, we anticipate that the presented methodology can be ultimately applied as a model-based downscaling tool for a higher spatial resolution of TWS, while comparing it to recently employed statistical downscaling [68,69].

**Author Contributions:** H.S.F. provided an initial concept, experimental design and critical comments concerning this research, and manuscript writing. Y.L. performed data processing, analyses, and interpretation.

**Funding:** This research was funded by the National Natural Science Foundation of China (NSFC) (Grant No. 41674007; 41429401; 41374010).

**Acknowledgments:** We appreciated GPS data product service platform of China Earthquake Administration for providing data support (http://www.cgps.ac.cn).

**Conflicts of Interest:** The authors declare no conflict of interest.

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
