# Peer review of "An Improved GPS-Inferred Seasonal Terrestrial Water Storage Using Terrain-Corrected Vertical Crustal Displacements Constrained by GRACE"

_remotesensing, doi:10.3390/rs11121433_

Round 1

Reviewer 1 Report

This paper investigates three approaches

\begin{itemize}
\item Tichonov regularization (TR)
\item  Helmert variance component estimation(HVCE)
\item  Akaike's Bayesian Information Criterion(ABIC)
\end{itemize}

to invert GPS vertical crustal displacement at 34 GPS sites to terrestrial water storage grid.\\

The authors describe TR and ABIC in section 4; but they do not explain HVCE.\\

My major comments are
\begin{itemize}
 \item  to describe HVCE in section 4; and \\

\item the quality of the figures should be improved, and they should be all consistent; authors should choose one between m, mm, and cm.
also in y axis use () instead of /, for example in figure 4 the y xis is Vertical Displacement (mm).
\end{itemize}

Here is the list technical comments:\\

\begin{enumerate}

\item page 5, fig.2: this figure has been cut off in half, and there is not any information about y-axis.\\

\item page 5, fig.3: what is y-axis? is it GPS hight or vertical displacement, in the caption by 'spectrum domain', you mean 'frequency domain'? \\

\item page 5, fig.4: this figure has been cut off in half also, and from the text, I understood that, you put a filter on fig.3 and the result is fig.4; if this is the case, fig.3 and fig.4 should be consistent in y axis, both in mm for example, and the y axis should be the same thing, either GPS height or vertical displacement. \\

\item page 4, line 167: the authors mention that FES2004 has been used for GPS data post-processing; They also should clarify if JPL GRACE Level-2 Release 05 (RL05) used the same ocean tidal loading for their data processing.  \\

\item page 4, line 140: what is GSM?\\

\item page 4, line 152: Is 'prediction/Oregon State University/Air force/Hydrologic' a web link?\\

\item page 6, please correct the reference [54] with the full authors name for Hector software package.\\

\item page 7, fig.5:  in the caption, mention clearly that a) is the annual amplitude and b) is the standard deviation.\\

\item page 7, line 239: what is TWS grid size? is it one degree by one degree?\\

\item page 8, $dS$ notation in the equation (4) is not the same as line 247, also in equation (9), $S$ is the Lagrangian target function which is confusing.  \\

\item page 8, there is not any information about the Tikhonov matrix $(L)$. \\

\item page 8, lines 266-275: this paragraph should be expand, giving more details about how many virtual GPS stations there are, are there only 7 virtual GPS stations as shown in figure 6. and give more explanation about how their location are chosen. Did authors considered to put  virtual GPS stations where the standard deviation of the annual amplitude is high from figure 5. \\

\item page 9, lines 285-292: it seems that lines 285-292  should be right after lines 266-275, and then in sepperate paragraph, before section 4
explain that the terrain correction has been applied to the GPS VCDs, with a clear formulation.\\

\item page 10, Table 1 has one extra row than the figure 6 for method; Figure 6 should be consistent with the Table 1.
also, in Figure 6, if authors show the difference between each method and GLDAS, it will be more useful.\\

\item page 12, figure 7: I do not understand figure 7. what are the blue dots in each graph.

\end{enumerate}

Author Response

To better view the equations and math symbols, the authors have uploaded the PDF version for your reading. Thank you.

This paper investigates three approaches 
- Tichonov regularization (TR)
- Helmert variance component estimation(HVCE)
- Akaike's Bayesian Information Criterion(ABIC)

to invert GPS vertical crustal displacement at 34 GPS sites to terrestrial water storage grid.
The authors describe TR and ABIC in section 4; but they do not explain HVCE.
Response: Yes, you are right. We add it into Section 4 in the revised manuscript as below:

".....While TR determines the value of k by obtaining the minimum of the trade-off curve between the model fitness and k, HVCE optimizes among the observations and constraints via their respective variance components iteratively. In this study, the equations for HVCE to determine the respective variance components can be written as [21]:

(7)

where ,

, , , , , ,

where  and  are the number of the observation and constraint equations, respectively;  and  are the estimated variance components for the observation and constraint equations to be solved iteratively, respectively;  and  are the column vectors of the observation and constraint residuals, respectively. By initializing a small value of k (i.e., 1×10-4), the value of k is iteratively determined via

,

(8)

until the  and are unchanged in the in the l-th iteration. "

My major comments are
- to describe HVCE in section 4; and 
- the quality of the figures should be improved, and they should be all consistent; authors should choose one between m, mm, and cm. also in y axis use () instead of /, for example in figure 4 the y xis is Vertical Displacement (mm).
Response: Given your comment, we added the description of HVCE in section 4 in the revised manuscript as shown in the above. We standardized the y-axis for the Vertical displacement (in mm) and terrestrial water storage [TWS] (in cm), respectively.

Here is the list technical comments:

1. page 5, fig.2: this figure has been cut off in half, and there is not any information about y-axis.

Response:

(i) We submitted a microsoft word '.doc' to the journal. It is apparent that the file was distorted when automatically converting into PDF in their journal online system. We will supply the undistorted PDF in the revised manuscript.

(ii) Yes, we added the caption in the y-axis of the revised figure.

2. page 5, fig.3: what is y-axis? is it GPS hight or vertical displacement, in the caption by 'spectrum domain', you mean 'frequency domain'?

Response: Yes, you are right - i.e., vertical displacement. It means filtering in frequency domain in the Figure caption. We fixed the Figure accordingly in the revised manuscript.

3. page 5, fig.4: this figure has been cut off in half also, and from the text, I understood that, you put a filter on fig.3 and the result is fig.4; if this is the case, fig.3 and fig.4 should be consistent in y axis, both in mm for example, and the y axis should be the same thing, either GPS height or vertical displacement.

Response: Indeed, after the filtering in the spectral domain (in Figure 3), we further applied the so-called "regional stacking filtering" to generate Figure 4. The reason for the regional stacking filtering described in line 191-196 of page 6 in the original manuscript.

Yes, we fixed the y-axis.

4. page 4, line 167: the authors mention that FES2004 has been used for GPS data post-processing; They also should clarify if JPL GRACE Level-2 Release 05 (RL05) used the same ocean tidal loading for their data processing. 

Response: No matter which data center (i.e., ITSG, JPL, CSR, GFZ, etc), GRACE Level-2 Release (RL05) did not correct for ocean tidal loading. The reason should probably be that the ocean tidal loading effect is significant only along the coastal regions [within 150 km away from the coasts only].

We clarify it by adding a sentence in the revised manuscript as below:

" Note that no ocean tidal loading correction is applied due to the study region far away from the coast." [line 155-156]

"....; (vi) to correct for the ocean tidal loading by choosing the FES2004 model [50] option in GAMIT [44], even though the study region is not close to the coast; " [line 190-191]

5. page 4, line 140: what is GSM?

Response: The GSM is a time series of GRACE Level-2 data products containing spherical harmonic coefficients (i.e., Stokes coefficients) that represents the mass variations on land up to degree 60/90/120 depending on the data processing centers (i.e., JPL, CSR, ITSG, GFZ, etc). Their provided data files are labeled as "GSM-2....."

Only a few reference papers try to interpret the full form of GSM terminology

GRACE model (GSM) (Schmeer et al., 2012)

GRACE Satellite-only Model (GSM) (Kanzow et al., 2005)

References:

Schmeer M., M. Schmidt, W. Bosch, F. Seitz: Separation of mass signals within GRACE monthly gravity field models by means of empirical orthogonal functions. Journal of Geodynamics, 59, 124132, 2012.

Kanzow, T., Flechtner, F., Chave, A., Schmidt, R., Schwintzer, P., & Send, U. (2005). Seasonal variation of ocean bottom pressure derived from Gravity Recovery and Climate Experiment (GRACE): Local validation and global patterns. Journal of Geophysical Research: Oceans, 110(C9).

6. page 4, line 152: Is 'prediction/Oregon State University/Air force/Hydrologic' a web link?

Response: National centers for environmental prediction/Oregon State University/Air force/Hydrologic research lab (NOAH) is one of the four land surface models within the Global Land Data Assimilation System (GLDAS). This can be downloaded at "https://mirador.gsfc.nasa.gov/".

We added the downloaded link and revised the sentences accordingly.

7. page 6, please correct the reference [54] with the full authors name for Hector software package.

Response: We corrected it in the revised manuscript.

8. page 7, fig.5:  in the caption, mention clearly that a) is the annual amplitude and b) is the standard deviation.

Response: We corrected the figure caption accordingly.

9. page 7, line 239: what is TWS grid size? is it one degree by one degree?

Response: The original TWS grid size is two degree by two degree (for JPL data products since it is expanded up to degree 90), but customarily interpolate into one degree by one degree before usage.

Note: In case of CSR data products expanded up to degree 60 only, the original TWS grid size is three degree by three degree (i.e, grid size = 180/degree).

We partly re-write section 2.3 and give the above description in the revised manuscript. We also change the sentences to "(i.e., the TWS of each 1°×1° grid in the study region with the extra 5° in each direction to be estimated)"

10. page 8, dS notation in the equation (4) is not the same as line 247, also in equation (9), S is the Lagrangian target function which is confusing. 

Response: Thank you for your careful checking. We corrected 'ds' to 'dS'. To avoid confusion, we changed  to  for the Lagrangian target function.

11. page 8, there is not any information about the Tikhonov matrix (L).

Response: We defined it in line 242-243 in the original manuscript as below:

"....where the Laplacian operator () is chosen as the Tikhonov matrix (L) for smoothing the solutions."

Given your comment, we place this information before the definition of design matrix A as below:

"... The Tikhonov matrix, L, is expressed in terms of Laplacian operator () for smoothing the solutions in this study. The design matrix, A, ...."

12. page 8, lines 266-275: this paragraph should be expand, giving more details about how many virtual GPS stations there are, are there only 7 virtual GPS stations as shown in figure 6. and give more explanation about how their location are chosen. Did authors considered to put virtual GPS stations where the standard deviation of the annual amplitude is high from figure 5.

Response: To give more explanation on the 7 virtual GPS stations chosen criteria, we revised manuscript regarding to line 266-275 as below:

"Due to the uneven distribution of the GPS stations with fewer stations near the boundary of the study region, poor GPS-inferred TWS is expected. To remedy the situation, GRACE VCD, serving as “virtual GPS stations”, are introduced at four locations close to the boundary of the study region that follows the southeast descending trend of the terrain with poor coverage of the GPS stations. Three more virtual GPS stations are chosen to compensate for the sparsely GPS situated at the steep slope (i.e., inclination angles of 31.1°) in the northwest of the study region (Figure 6). Note that the GRACE VCD can be converted by the forward modeling formulation in [11] using GRACE-inferred TWS, in which the load Love numbers are adopted from [43]. The annual amplitude of the GRACE VCD are subsequently synthesized using Equation (1) for the pre-defined virtual GPS stations based on the above criteria. The chosen virtual GPS stations should be as few as possible, such that the GPS can dominate the inversion process. This strategy is equivalent to the introduction of stochastic constraints into the linear observation model [67]. "

By that time, we did not consider to put virtual GPS stations where the standard deviation of the annual amplitude is high from figure 5, but consider removing the GPS stations with high standard deviation. Overall, your idea certainly deserves for further investigation in the future.

13. page 9, lines 285-292: it seems that lines 285-292  should be right after lines 266-275, and then in sepperate paragraph, before section 4 explain that the terrain correction has been applied to the GPS VCDs, with a clear formulation.

Response: Thank you for your suggestion. It helps the logic flow. We did it accordingly in the revised manuscript.

14. page 10, Table 1 has one extra row than the figure 6 for method; Figure 6 should be consistent with the Table 1. also, in Figure 6, if authors show the difference between each method and GLDAS, it will be more useful.

Response: The one extra row is indeed the spatial plots of the annual amplitude of TWS inferred from GRACE and TWS-modeled GLDAS. We add a new plot in figure 7 [new], in order not to be confusing.

As per your request, we generate another spatial plot (Figure 8 [new]) showing bias difference between each method and GLDAS in the revised manuscript.

15. page 12, figure 7: I do not understand figure 7. what are the blue dots in each graph.

Response: The blue dots are the 64 gridded TWS of 1°×1°in the study region plotting as phase diagram, that corresponds our derived TWS to GLDAS-modeled TWS. The purpose of this original Figure 7 is to show the bias. As for your suggestion in the above question 14, we replace it with the Figure 8 [new] in the revised manuscript.

Reviewer 2 Report

The paper discusses an experiment to infer Terrestrial water storage by means of GPS displacement and GOCE data.

The topic is of broad interest for the scientific community, however in my opinion the paper should be better refined before a possible publication in remote sensing. In the following the authors can find some points in order to improve their work:

1)      I suggest to avoid the use of acronyms in the abstract.

2)      Section 2.3. It is not clear to me what is the GRACE dataset used in the paper. Do you use GRACE TWS (or EWH)? In case how do you transforms GRACE TWS on VCD? Please add a sentence on this points. Also please add the correct acknowledgment and citation as found in the website.

3)      Section 3. The authors put GPS-preprocessing and GPS post-processing Sections. It seems that the GPS processing is missing. I suggest to change GPS-preprocessing in GPS processing.

4)      Section 3.1. It is not clear to me what kind of GPS preprocessing has be done. It is a network solution with the IGS stations? What is the frequency of the solution (e.g. you compute ppt solutions, daily solutions,…). Please give all the details on the GPS preprocessing.

5)      Section 3.2 I guess that the annual period can be difficulty estimate from the used dataset (I think that 2.5 years is a too short period in order to properly recover information on such 1 cpy signal, especially in presence of coloured noise). Can you please add a discussion on this point?

6)      Figure 2. Part of the figure is missing (please check and fix the problem). Please add units in the axis.

7)      Figure 3. Can you please enlarge the font size?

8)      Please add a reference on the Hector software package an on the specific algorithm you used.

9)      Eq. 2 Is there any assumption required to apply Eq.2? in case can you please add a discussion on the required assumptions?

10)   Figure 6. Reference letters on the figure are missing.

Author Response

The paper discusses an experiment to infer Terrestrial water storage by means of GPS displacement and GOCE data.

The topic is of broad interest for the scientific community, however in my opinion the paper should be better refined before a possible publication in remote sensing. In the following the authors can find some points in order to improve their work:

1)      I suggest to avoid the use of acronyms in the abstract.

Response: Yes, we removed all acronyms in the abstract.

2)      Section 2.3. It is not clear to me what is the GRACE dataset used in the paper. Do you use GRACE TWS (or EWH)? In case how do you transforms GRACE TWS on VCD? Please add a sentence on this points. Also please add the correct acknowledgment and citation as found in the website.

Response: Our original aim is to simplify the data description, but the writing created confusion and misunderstanding of the data usage. We admitted that we were not writing well in this section. Following your suggestion, we partly re-write it with more descriptions in the revised manuscript with related references as below:

"GRACE is a satellite mission project of NASA and German Aerospace Center (DLR) measuring time-variable Earth's gravity changes that can be converted into TWS variations on a global scale [4]. The degree-90 GRACE Level-2 Release 05 (RL05) GSM monthly gravity data product, released by JPL, contains spherical harmonic coefficients (i.e., Stokes coefficients) that represents the mass variations [39,40]. These data are freely available at https://grace.jpl.nasa.gov/. This product allows us to calculate the time series of TWS (in terms of EWH in cm at a regular grid) using equation (14) in [7] divided by the water density. Gaussian filter at a radius of 300 km [41] and DDK decorrelation filter [42]are applied to reduce the spatially correlated error of TWS data at higher degrees, with final gridded TWS at 1°×1° grid. Subsequently, GRACE TWS are converted into GRACE VCD by the forward modeling formulation in [11], in which the load Love numbers are adopted from [43]. This is followed by introducing GRACE VCD at specified locations as "virtual GPS stations" serving as spatial data constraints for the inversion process..." [line 142-155]

References:

4. Tapley, B.D.; Bettadpur, S.; Watkins, M.; Reigber, C. The gravity recovery and climate experiment: Mission overview and early results. Geophys. Res. Lett. 2004, 31,

7. Wahr, J.; Molenaar, M.; Bryan, F. Time variability of the Earth's gravity field: Hydrological and oceanic effects and their possible detection using GRACE. J Geophys. Res. Solid Earth 1998, 103(B12), 30205–30229.

11. Van Dam, T.; Wahr, J.; Lavallée, D. A comparison of annual vertical crustal displacements from GPS and Gravity Recovery and Climate Experiment (GRACE) over Europe. J. Geophys. Res. 2007, 112(B3).

39. Swenson, S. GRACE Monthly Land Water Mass Grids NETCDF Release 5.0. Ver. 5.0. PO. DAAC, CA, USA. 2012.

40. Landerer, F.W.; Swenson, S.C. Accuracy of scaled GRACE terrestrial water storage estimates. Water Resour. Res. 2012, 48, 4531.

41. Swenson, S.; Wahr, J. Post-processing removal of correlated errors in GRACE data. Geophys. Res. Lett. 2006, 33.

42. Kusche, J.; Schmidt, R.; Petrovic, S.; Rietbroek, R. Decorrelated grace time-variable gravity solutions by gfz, and their validation using a hydrological model. J. Geod. 2009, 83(10), 903–913.

43. Guo, J.Y.; Li, Y.B.; Huang, Y.; Deng, H.T.; Xu, S.Q.; Ning, J.S. Green’s Function of Earth’s Deformation as a Result of Atmospheric Loading. Geophys. J. Int. 2004, 159, 53–68.

3)      Section 3. The authors put GPS-preprocessing and GPS post-processing Sections. It seems that the GPS processing is missing. I suggest to change GPS-preprocessing in GPS processing.

Response: We admit that we were not writing well in this section. We re-write the whole section with more details according to your request, as responded in your comment 4 and 5 below.

4)      Section 3.1. It is not clear to me what kind of GPS preprocessing has be done. It is a network solution with the IGS stations? What is the frequency of the solution (e.g. you compute ppt solutions, daily solutions,…). Please give all the details on the GPS preprocessing.

Response: According to your request, we re-write this section in the revised manuscript as below:

"The GAMIT version 10.4 software [44] was employed for preprocessing the CMONOC GPS observation data to generate daily solution of station coordinates and their uncertainties. By processing 34 CMONOC GPS stations together with 24 International GNSS Service (IGS) [45] stations surrounding China, the network solution for the coordinates of the 34 CMONOC GPS station time series were aligned to the International Terrestrial Reference Frame 2008 (ITRF2008) [46]. The uncertainties of the 34 CMONOC GPS station coordinates were initially constrained to 100 m by using approximate coordinates from point-position solution, whereas that of the 24 IGS station coordinates were constrained to 5 cm. The Earth Orientation Parameters (EOP) were also constrained to priori values listed in International Earth Rotation Service (IERS) Bulletin B.

During the preprocessing, corrections were applied in GAMIT to remove systematic effects . The following correction procedures were applied: (i) to fix precise orbits by the IGS final ephemeris products; (ii) to correct for the ionospheric delay up to three order terms [47] in GAMIT; (iii) to correct for the tropospheric delay by using the Vienna mapping function 1 (VMF1) [48] with a cut-off values of 10° and a priori hydrostatic delays provided by the global pressure and temperature (GPT) model [49]; (iv) to correct for the receiver antenna offsets using the IGS antenna correction files; (v) to correct for the non-tidal atmospheric loading by MIT atmdisp_cm.year files;  (vi) to correct for the ocean tidal loading by choosing the FES2004 model [50] option in GAMIT [44], even though the study region is not close to the coast; (vii) to correct for the solid Earth tides and pole tide according to the IERS standard [51].

Environmental loading includes atmospheric, hydrological, and non-tidal ocean loading (NTOL) [52]. Since hydrological loading signal is our main concern, other environmental loading signals in the GPS VCD time series should be corrected. Note that the atmospheric loading has been corrected in the aforementioned preprocessing. Therefore, GPS VCD induced by NTOL should be corrected to obtain purely hydrological loading signal due to TWS. This can be corrected by using the half-daily modeled NTOL displacement data with 2.5°×2.5° spatial resolution obtained from Global Geophysical Fluid Center (GGFC), in which these data are available at http://geophy.uni.lu/." [line 170-198]

Reference list:

44. Herring, T.; King, R.; McClusky, S. GAMIT reference manual, release 10.4. Massachusetts Institute of Technology, Cambridge 2010.

45. Dow, J.M.; Neilan, R.E.; Rizos, C. The international GNSS service in a changing landscape of global navigation satellite systems. J. Geod.2009, 83, 191–198.

46. Altamimi, Z.; Collilieux, X.; Métivier, L. ITRF2008: An improved solution of the international terrestrial reference frame. J. Geod.2011, 85, 457–473.

47. Petrie, E.J.; King, M.A.; Moore, P.; Lavallée, D.A. Higher‐order ionospheric effects on the GPS reference frame and velocities. J. Geophys. Res. Solid Earth2010, 115.

48. Boehm, J.; Werl, B.; Schuh, H. Troposphere Mapping Functions for GPS and Very Long Baseline Interferometry from European Centre for Medium-Range Weather Forecasts Operational Analysis Data. J. Geophys. Res. 2006, 111.

49. Boehm, J.; Heinkelmann, R.; Schuh, H. Short note: A global model of pressure and temperature for geodetic applications. J. Geod.2007, 81, 679–683.

50. Lyard, F.; Lefevre, F.; Letellier, T.; Francis, O. Modelling the global ocean tides: modern insights from FES2004. Ocean dyn 2006, 56(5-6), 394415.

51. Petit, G.; Luzum, B. IERS conventions (2010); BUREAU INTERNATIONAL DES POIDS ET MESURES SEVRES (FRANCE): 2010.

52. Jiang, W.; Li, Z.; van Dam, T.; Ding, W. Comparative analysis of different environmental loading methods and their impacts on the GPS height time series. J. Geod.2013, 87, 687–703.

5)      Section 3.2 I guess that the annual period can be difficulty estimate from the used dataset (I think that 2.5 years is a too short period in order to properly recover information on such 1 cpy signal, especially in presence of coloured noise). Can you please add a discussion on this point?

Response: Yes, we agree. We add a small discussion on this point as below:

" Due to the presence of the colored noise, the nearly 3-year data span is too short for a proper recovery of annual signal. The traditional least-squares method will also lead to a biased estimation [61]. Consequently, maximum likelihood estimation was employed to estimate all unknown parameters in Equation (1), as the colored noise models can be assumed and estimated in order to better recover the annual signal. " [line 248-252]

6)      Figure 2. Part of the figure is missing (please check and fix the problem). Please add units in the axis.

Response: We submitted a microsoft word '.doc' to the journal. It is apparent that the file was distorted when automatically converting into PDF in their journal online system. We will supply the undistorted PDF in the revised manuscript and fix the axis unit, etc.

7)      Figure 3. Can you please enlarge the font size?

Response: Yes, we did accordingly.

8)      Please add a reference on the Hector software package an on the specific algorithm you used.

Response: Yes, we add a reference on the Hector software package. The employed algorithm for noise model is the combination of power-law noise and white noise, which is an option to be chosen in the software package. The sentences are revised as below:

" Hector software package [62] employing maximum likelihood estimation was used to determine all parameters. The power-law noise and white noise combination were chosen as the noise model algorithm within the Hector software package option." [line 252-255]

9)      Eq. 2 Is there any assumption required to apply Eq.2? in case can you please add a discussion on the required assumptions?

Response: We state the required assumptions in the introduction in line 77-78, where the discussion is in line 78-81. To be clear, we also rephrase the sentences in the beginning of section 4 as below:

" The field displacement at a point, as observed from a GPS station, represents the response of all loading masses contribution on the rigid plane surface with elasticity assumption [14]. "

10)   Figure 6. Reference letters on the figure are missing.

Response: Perhaps the letters are not apparent. It was indeed in the top-right corner of each sub-plots. In the revised manuscript, we enlarged the reference letters and placed it in a brighter region.

Round 2

Reviewer 2 Report

All my previous comments have been properly adressed by the authors. The paper can be in my opinion published on Remote Sensing.